# Heat Acclimation with or without Normobaric Hypoxia Exposure Leads to Similar Improvements in Endurance Performance in the Heat

**DOI:** 10.3390/sports10050069

**Published:** 2022-04-30

**Authors:** Erik D. Hanson, Matthew B. Cooke, Mitchell J. Anderson, Tracey Gerber, Jessica A. Danaher, Christos G. Stathis

**Affiliations:** 1Department of Exercise and Sport Science, University of North Carolina at Chapel Hill, Chapel Hill, NC 27599, USA; edhanson@email.unc.edu; 2Institute for Health and Sport, Victoria University, Melbourne, VIC 8001, Australia; 3Department of Health Sciences and Biostatistics, Swinburne University, Melbourne, VIC 8001, Australia; mbcooke@swin.edu.au; 4Metabolic and Vascular Physiology, Baker IDI Heart and Diabetes Institute, Melbourne, VIC 3004, Australia; mitch@shinbonemedical.com; 5College of Health and Biomedicine, Victoria University, Melbourne, VIC 8001, Australia; tracey.gerber@allg.org.au (T.G.); jessica.danaher@rmit.edu.au (J.A.D.); 6School of Science, STEM College, Royal Melbourne Institute of Technology (RMIT), Melbourne, VIC 3001, Australia

**Keywords:** heat acclimation, plasma volume, hypoxia, concurrent training, altitude training

## Abstract

Background: Combining the key adaptation of plasma volume (PV) expansion with synergistic physiological effects of other acclimation interventions to maximise endurance performance in the heat has potential. The current study investigated the effects of heat acclimation alone (H), combined with normobaric hypoxia exposure (H+NH), on endurance athletic performance. Methods: Well-trained participants completed a heat-stress trial (30 °C, 80% relative humidity (RH), 20.8% fraction of inspired oxygen (FiO_2_)) of a 75 min steady-state cycling (fixed workload) and a subsequent 15 min cycling time trial for distance before and after intervention. Participants completed 12 consecutive indoor training days with either heat acclimation (H; 60 min·day^−1^, 30 °C, 80% RH; 20.8% FiO_2_) or heat acclimation and overnight hypoxic environment (H+NH; ~12 h, 60% RH; 16% FiO_2_ simulating altitude of ~2500 m). Control (CON) group trained outdoors with average maximum daily temperature of 16.5 °C and 60% RH. Results: Both H and H+NH significantly improved time trial cycling distance by ~5.5% compared to CON, with no difference between environmental exposures. PV increased (+3.8%) and decreased (−4.1%) following H and H+NH, respectively, whereas haemoglobin concentration decreased (−2%) and increased (+3%) in H and H+NH, respectively. Conclusion: Our results show that despite contrasting physiological adaptations to different environmental acclimation protocols, heat acclimation with or without hypoxic exposure demonstrated similar improvements in short-duration exercise performance in a hot environment.

## 1. Introduction

Athletes are constantly aiming to improve athletic preparation for events, overcoming environmental challenges and seeking natural and legal training assistance to facilitate physiological adaptations and enhance their performance. It is well established that athletic performance is often decreased in hot, humid environments [1,2] and at altitude [3]. Heat acclimation and hypoxic exposure (simulated or real) are common strategies used prior to competition to help attenuate reductions in performance under hot, humid conditions [4], and enhance performance at sea level or events contested at altitude [5,6], respectively.

Heat-acclimation protocols vary [7], but typically involve frequent and repeated exposure to exercise heat-stress that results in elevations of core and skin temperatures to a level that challenges thermoeffector responses [8,9,10]. There are several classical physiological metrics that indicate adaptation to the heat, including plasma volume (PV) expansion within ~3 days; resulting in lower heart rate (HR), increased stroke volume and for a given work rate, maximal cardiac output [11,12,13,14]. Heat-induced hypervolemia can lead to an increase in total specific heat capacity of the blood [15], and improved sweat rates and reduced electrolyte losses in response to exercise, though the latter is usually complete after ~10 days [11,13]. Metabolic adaptations such as reduced muscle glycogen depletion and lower exercise muscle and blood lactate accumulation are also evident, especially during or immediately after sub-maximal exercise [16,17]. Ultimately, these adaptive changes are designed to improve cardiovascular stability and reduce physiological and metabolic stress [14,18,19] during subsequent heat exposure while competing, and thus translate into improved performance [20]. The ‘Live High-Train Low’ (LHTL) training strategy is where athletes adapt to hypoxia by living and sleeping at high altitude (real or simulated), but train at sea level to avoid the hypoxia-induced reduction in maximal training intensity [21,22]. The potential underlying mechanisms for improved exercise performance following LHTL include an increase in erythropoiesis, blood-oxygen transporting capacity and enhanced muscle buffering capacity and mitochondrial efficiency [23,24,25]. These adaptations, with or without an increase in maximal oxygen consumption (VO_2_max), can lead to enhanced exercise performance at sea level [24,26].

The time course for adaptations following heat and hypoxic exposure appear to differ, with slightly longer exposure times required for hypoxia compared to heat. Heat acclimation/acclimatisation is relatively rapid, with a significant fraction of the heat-induced adaptations occurring during the first week of exposure, and 10–14 days required for complete or near-complete acclimation/acclimatisation [11,27]. For those undertaking LHTL, total hypoxic exposure dose is an important consideration, since daily exposure is less than those that undertake both living and training at altitude. Adaptations to LHTL require about 3–4 weeks at a simulated altitude of 1800–3000 m with an average of 14 h of exposure per day [24,28]. However, improvements in both haematological and non- haematological adaptations have been noted following short exposure times [23], though whether this translates into improved performance requires further investigation.

The theoretical combining of the aforementioned major physiological adaptations through the concurrent use of heat and altitude training methods has been a topic of recent discussion to further enhance performance. However, at present, there are a limited number of studies examining the cross-acclimation benefits of combining heat acclimation and ‘LHTL’ model, and current findings are equivocal. Team sports players exposed to both environmental stimuli during a preseason camp displayed improved Yo-Yo Intermittent Recovery test level 2 (Yo-YoIR2) test performance and increased PV, with no difference between environmental exposures [29]. Additionally, in males, improvements in maximal oxygen uptake, total work performed and/or lactate threshold were similar following both environmental conditions [8,30]. Conversely, in well-trained runners, heat-only exposure improved performance (3 km time trial), with no effect after heat and hypoxia exposure combined [31]. However, differences in study design, such as lack of a control group [29], total duration of heat [31] and hypoxic exposure [8] and the use of untrained individuals [30], make it difficult to draw any definitive conclusions. Further, reported changes in PV expansion, a key component of heat acclimation, is often equivocal following combined environmental exposures [8,29,30,31].

Given the majority of primary performance outcomes are typically assessed in temperate environments [8,29,31], additional studies are warranted to confirm whether the concurrent actions of these two established methods can further enhance endurance performance in hot, humid conditions in trained athletes. Therefore, the purpose of this study was to investigate the combined environmental exposure of heat acclimation with or without hypoxia on physiological and metabolic variables during steady-state and maximal-effort performance in the heat. We hypothesised that 12 days of daily heat acclimation with hypoxic altitude simulation enhances cycling performance to a greater extent than heat acclimation alone in well-trained athletes.

## 2. Materials and Methods

### 2.1. Participants and Design

Twenty-one well-trained male and female endurance athletes (cyclists and triathletes) participated in this study. Well-trained was defined as those with a VO_2_max > 55 mL·kg·min^−1^ and training consistently for >4 years, averaging 4–5 days per week (>7 h per week). All participants were from the local area and had been training in cool winter climates (maximum daily temperature during study 16.5 ± 0.8 °C) near sea level (elevation, 31 m) in preparation for endurance events in hot, humid climates. Participants were divided into three groups: a group that underwent a heat-acclimation process (H, N = 7), a group that underwent heat acclimation and slept in a simulated normobaric altitude environment (H+NH, N = 7) and a control group that did not undergo any acclimation protocols but kept training per their normal routine (CON, N = 7). Participants were allocated to groups prior to baseline testing on the basis of their ability to attend the required sessions, as the H and H+NH groups were required to train daily in a heat chamber and the H+NH also needed to be available to remain at the testing facility overnight. Given the open-label design of the study, all participants were aware of their group allocation. All participants continued their normal training in addition to the study requirements for the 2 weeks. The CON group were instructed to complete an extra 60 min session at a similar intensity to match the 60 min heat-acclimation session undertaken by the H and H+NH groups. Prior to baseline testing, all participants began an iron supplementation program of 525 mg·day^−1^ (Ferro-Gradument^®^, Abbott Australiasia, Botany, NSW, Australia) that continued throughout the duration of the study to ensure adequate levels for adaptation to altitude [22]. All participants completed VO_2_max testing and heat-stress trials before and at the end of the study. All procedures were submitted to and approved by the Victoria University Human Research Ethics Committee (HRETH 11/149, approval date 30 August 2011).

### 2.2. Graded Exercise Test

All participants were assessed for VO_2_max on an electronically braked cycle ergometer (Velotron Pro, RacerMate Inc., Seattle, WA, USA) to determine cardiovascular endurance under laboratory conditions (18–20 °C, 30–50% relative humidity (RH)). Females initially completed stages at 100 Watts (W) and 150 W, whereas males completed stages at 150 W and 200 W, before progressing by 25 W until volitional fatigue. All stages were 2.5 min in length. Peak wattage was calculated as a percentage of the wattage attempted during the final stage of the test along with the time completed at that stage, as used previously [32]. Pulmonary ventilation and expired gas concentrations were analysed in real time using an automated computerised indirect calorimetry system (Ametek S-3A/II and Ametek CD-3A, Pittsburgh, PA, USA). The system was calibrated daily and between each test. Oxygen consumption (VO_2_) was considered maximum if a plateau was achieved (VO_2_ increase of <150 mL·min^−1^ with increased work). In the absence of a clear plateau, tests were still considered maximal efforts if at least two of the following secondary criteria were met: a respiratory exchange ratio (RER) of > 1.10, a rating of perceived exertion (RPE) > 18, and a peak HR within 10 beats·min^−1^ of the age-predicted maximum (220-age). Heart rates were recorded using portable monitors (Polar Electro Inc., Lake Success, NY, USA) throughout the test.

### 2.3. Heat-Stress Trial

Approximately one week after the VO_2_max test, all participants completed the baseline heat-stress trial. The 90 min exercise trial, performed on an electronically braked cycle ergometer (Velotron Pro, RacerMate Inc., Seattle, WA, USA), was divided into two sections: (1) 75 min steady-state effort at 63% of peak power output (PPO) achieved during the VO_2_max test and (2) 15 min maximal-effort time trial for distance performed after 2 min rest following the steady-state effort. The percentage determined for PPO was based on a corresponding estimated lactate threshold of ~95%, according to previous work [33]. This trial was designed as a controlled race simulation of a defined workload phase with a high-intensity time-trial finish. The trial was performed in environmental conditions of 30 °C and 80% RH with fans to recirculate air, but specifically not directed at the participants.

Participants arrived in the laboratory after an overnight fast. Baseline body mass was collected prior to an IV cannula being placed in a forearm vein for repeat blood sampling. Baseline blood samples were collected after 10 min of supine rest. Resting respiratory gases and tympanic temperature (Braun ThermoScan Pro 4000, Welch Allyn, Skaneateles Falls, NY, USA) were measured in duplicate (values within 0.1 °C) and collected following the baseline blood sample. Participants then entered the environmental chamber and commenced exercise. Participants consumed a standardised quantity (4 mL·kg bodyweight^−1^) of sports drink (Electrolyte Replacement Formula, Musashi, Notting Hill, Melbourne, Vic, Australia) every 15 min throughout the trial and had ad libitum access to water. Blood samples, tympanic temperature, HR, RPE, VO_2_, and RER were obtained at regular intervals throughout exercise (Figure 1). Immediately following the time trial, participants recovered in the supine position at ambient temperatures (~20 °C) and additional blood samples were obtained.

### 2.4. Carbon Monoxide (CO)-Rebreathing Method

The present study had planned to use the optimised CO-rebreathing technique based on Schmidt and Prommer, 2005 [34]. Briefly, this method requires inhaling and rebreathing a bolus of CO through a spirometer for 2 min and analysis of the increase in the carboxyhaemoglobin (COHb) content of capillary blood at about 7 min after inhalation of CO. The COHb levels would be analysed by an independent pathology lab. Unfortunately, we identified an issue with the haematological values when provided by the pathology lab. Moreover, this was not until 3–4 days after the intervention had started. Thus, given the lack of confidence in the baseline values, the study having started and as we were unable to resolve the issue quickly, we made the decision to not utilise the CO-rebreathing method post intervention. We think it important to highlight our issues for future researchers that are testing large number of participants in a short period of time to ensure their values are correct before starting the intervention.

### 2.5. Blood Measurements

Blood samples obtained at rest, during exercise and in recovery were immediately centrifuged (12,000 rpm for 2 min at room temp), with the plasma extracted and frozen in liquid nitrogen for future analysis. Plasma lactate levels were assessed in duplicate using an automated analyser (YSI 2300 STAT Plus, YSI Inc., Yellow Springs, OH, USA). Resting full blood examination profiles were completed before and after acclimation using certified pathology services (Dorevitch Pathology, Heidelberg West, Melbourne, VIC, Australia, Accredited Laboratory #2204) to screen for anaemia, infection or any electrolyte abnormalities. Haematocrit (Hct) and Hb values from before and after acclimation were used to estimate PV changes [35]. Sweat rates were estimated by the change in body mass after accounting for total fluid volume consumed over the total exercise time (90 min).

### 2.6. Heat-Acclimation Protocol

Heat-acclimation sessions for the H and H+NH groups commenced on the day following the heat-stress trial. Participants completed 12 consecutive sessions that were 60 min in length at a constant workload of ~58% of PPO, determined by the baseline VO_2_max test at the same environmental conditions as the heat-stress trial. The initial daily workloads were adjusted slightly as required for the participants to complete the sessions. For the participants preparing for upcoming competitions, they completed one long weekend training ride (3+ h) during the experiment, and the training load for the heat-acclimation session was reduced for that single session only to ensure 60 min of heat exposure was still accumulated. After each heat-acclimation session, the participants consumed a 30 g protein recovery drink (Musashi, Notting Hill, Melbourne, Vic, Australia). Tympanic temperature was assessed before, during and after each session, whereas RPE was assessed during the session.

### 2.7. Simulated Altitude Environment

In addition to the heat-acclimation sessions, the H+NH group spent ~12 h per night at a simulated altitude of 2500 m in normobaric conditions via oxygen displacement with nitrogen (60% RH; 16% FiO_2_; ‘altitude hotel’, Kinetic Performance Technology, Mitchell, ACT, Australia). The room dimensions were as follows: kitchen/common area 36.8 m^2^; bedrooms ranged from 13.8 to 14.7 m^2^, with up to 4 sleeping in one room; bathroom, toilet and shower ranged from 3.7–2.3 m^2^. Oxygen saturation and HR (Nellcor™ N-600x Pulse Oximetry Monitor, Covidien, Dublin, Ireland) were measured after the first hour of hypoxia and in the morning before departing the facility to ensure participants’ safety. The H+NH participants entered the altitude hotel in the evening (approx. 6 p.m.) and remained in the hotel until morning (approx. 6 a.m.). The H+NH participants consumed their evening and morning meals within the hypoxic environment. While participants were free to walk around, sleep was the primary activity within the altitude hotel and hypoxic conditions were monitored every four hours and remained constant throughout the study.

### 2.8. External Self-Training Exposure

All participants in the study kept a training diary to monitor all additional training performed outside the context of the study. The total training load was quantified in average minutes and was the sum of all activities (i.e., swimming, running, cycling, gym work) performed across the duration of the study. Self-reported intensity or pace was also recorded.

### 2.9. Statistical Analysis

All statistical analyses were performed using SPSS v25 (IBM Corp., Armonk, NY, USA). Differences between all groups (Three-Group Model) were determined using mixed model, repeated measures ANOVA with a Tukey post hoc analysis. Pre-planned comparisons between H+A and H only (Two-Group Model) were also analysed using the same approach. In the event of a significant interaction, simple effects were used to determine where the differences occurred. Effect sizes were calculated as Cohen’s d (*d*) values of 0.2, 0.5, and 0.8 indicating small, medium and large effect sizes, respectively. Statistical significance was set at *p* < 0.05. Data are reported as mean ± SEM.

## 3. Results

### 3.1. Participant Characteristics

The participants in this study recorded a collective VO_2_max average of 60.7 mL·kg·min^−1^ with no differences between the groups for either absolute or relative VO_2_max, VO_2_max peak watts, and trial workloads at baseline (Table 1). There was no significant difference in total self-training exposure external to the intervention training (*p* = 0.474, Table 2), modality of training (Swim *p* = 0.379, Bike *p* = 0.574, Run *p* = 0.616 and other *p*= 754, Table 2) or respective exercise modality intensity (Swim *p* = 0.690, Bike *p* = 0.873, and Run *p* = 0.424, Table 2).

### 3.2. Pre-Intervention (Baseline) Heat Trial—Physiological Responses

The physiological responses to the initial (baseline) heat trial between groups is displayed in Table 3. There were no group differences in the response to the steady state or maximal effort portions of the trial. VO_2_ and RER significantly increased initially with exercise and remained constant during steady state (*p* < 0.05). Initially, temperature, HR and lactate levels significantly increased during the trial (*p* < 0.05) and they peaked following the maximal effort time trial (*p* < 0.01).

### 3.3. Body Mass before and after Intervention and Heat Trial

Body mass measured before and after the interventions and heat trial are displayed in Table 4. A significant group × trial interaction for body mass was identified when comparing all three groups (Three-Group Model, *p* = 0.049) and when comparing H+NH and H only (Two-Group Model, *p* = 0.02). All groups lost body mass during one or both heat trials (*p* = 0.002). However, the H+NH group lost on average 1.3 kg of body mass (*d* = 0.22) at the end of the 12 days, while the other groups remained weight stable.

### 3.4. Haematological Responses to Interventions

Blood parameters before and after interventions are displayed in Figure 2. A significant group × time interaction between H and H+NH was identified for (Hb) (Two-Group Model, *p* = 0.043), with H+NH increasing (Hb) by ~2.9% (*d =* 0.38) and H decreasing (Hb) by ~2% (*d =* 0.31). The calculated plasma-volume changes demonstrated a trend (*p* = 0.081) for opposing acclimation response, with H increasing by 3.8% (*d =* 0.65) and H+NH decreasing by −4.1% (*d =* 0.78).

### 3.5. Post-Intervention Heat Trial—Physiological Responses during Sub-Maximal Steady State Effort

Several of the key physiological adaptations were examined in the three-group model (H, H+NH, and CON) and the Two-Group Model (H vs. H+NH) during steady-state exercise. The Three-Group Model identified a trend for a significant trial × group interaction for HR (Figure 3A, *p* = 0.081), but this was no longer apparent when comparing H and H+NH only (*p* = 0.278). There were no differences in lactate concentrations between groups or across trials using both Three- and Two-Group Models (Figure 3B).

A trend for a trial × group interaction was observed for changes in tympanic temperature following exposure in the Three-Group Model (Figure 3C, *p* = 0.069). In the Two-Group Model (*p* = 0.025), the H group demonstrated significantly lower temperatures across the duration of the trial (average: −1.1 °C, 2.9%, *d =* 0.68, *p* < 0.01) compared to the H+NH group (*d =* 0.08).

### 3.6. Post-Intervention Heat Trial—Physiological Responses to Maximal-Effort Time-Trial for Distance

Following heat acclimation with and without hypoxia exposure, there was a trend for significance in time-trial distance (*p* = 0.059 for group × time interaction), with further distance covered in the H (*d =* 1.27) and H+NH (*d =* 0.79) groups, and no change in the control group. When examining the Two-Group Model, time-trial distance was significantly improved (+5.5%) from baseline to post-intervention testing (Figure 4A, *p* = 0.016), with no difference in performance between groups. At baseline, there were no differences in peak lactate levels between groups after the time trial, but following interventions, H+NH and H only were 33% (*d =* 0.98) and 18% (*d =* 0.57) higher, respectively, compared to CON (group × trial interaction, *p* = 0.052, Figure 4B). When examining the Two-Group Model, peak lactate levels were significantly higher from baseline to post-intervention testing (Figure 4B, *p* = 0.02), with still no difference between groups.

## 4. Discussion

In the present study, heat acclimation improved ‘race simulated’ time-trial performance following steady-state exercise in hot humid conditions; however, superimposing normobaric hypoxic exposure did not further augment performance in well-trained athletes. Physiological indicators of improved thermoregulation were mixed within groups, with only trends typically observed and no further benefits of hypoxia noted. However, caution is needed when interpreting these changes, given limitations in the haematological methods used in the present study. A reduction in body mass following the combined heat acclimation and hypoxia was observed, which potentially indicates an altered fluid shift and consequential impact on circulating fluid volume compared with the heat acclimation only. These results suggest that despite possible contrasting adaptations in some key haematological measures with the addition of hypoxia to heat acclimation, short-duration exercise performance is still improved in hot, humid conditions.

Twelve days of daily heat exposure with or without overnight hypoxic exposure increased cycling time-trial distance by an average of 5.5% compared to controls, with no difference between H and H+NH interventions. At the time the present study was designed (late 2011), this finding was contrary to our hypothesis, which anticipated that combining heat with hypoxia would result in greater physiological benefits for improved performance in hot environments. However, additional studies published since then appear to support our current findings of no added benefit when such methods are combined [8,31]. Buchheit et al. [29] demonstrated improvements in the Yo-YoIR2 (44%) under temperate conditions (23 °C, normal air) directly after 12–14 days of heat with or without hypoxic exposure, with no difference between treatments. Likewise, Rendell and colleagues [8] demonstrated improvements, albeit less (+4%), in a 30 min work-performed cycling trial under temperate conditions (22 °C, 50% RH) immediately following heat acclimation with or without overnight normobaric hypoxia. However, these ergogenic benefits were no longer apparent in either group following a 2-week retention period. Conversely, using magnitude-based inferences, McCleave et al. [31] showed likely improvement in temperate–normoxic 3 km running trial performance (3.3%) at 3 weeks (but not immediately) following 3 weeks of heat acclimation only, with no ergogenic benefit when combined with hypoxia.

While both Buchheit et al. [29] and Rendell et al.’s [8] studies showed similar improvements between intervention groups, given the absence of a control group, it is difficult to draw direct comparisons to the present study regarding the ergogenic benefits of each exposure interventions compared to no treatment at all. In the current study, both exposure interventions improved time-based cycling performance compared to control. This is contrary to McCleave et al. [31], whose study showed improvement only after heat acclimation and not heat plus hypoxia or control (live at <600 m and trained at 14 °C, 55% RH). This lack of improvement in the heat and hypoxia group was supported by limited thermoregulatory adaptations (except for Hbmass) compared to independent heat training [31,36]. Comparing the two studies, McCleave et al. [31] used participants of similar fitness levels; hypoxic exposure was slightly higher in altitude (~3000 m, 13 h·day^−1^ vs. ~2500 m, ~12 h·day^−1^) and longer (3 weeks vs. 2 weeks); and heat exposure was different (three times a week vs. daily). However, despite the intermittent nature of heat exposure in the McCleave study, evidence of heat adaptation in key haematological markers (i.e., an increase in PV) were observed [31]. A major difference between studies was the environmental conditions in which the performance test was undertaken. Our study tested under hot, humid conditions, while McCleave et al. [31] tested under temperate conditions. The physiological explanation for this is not readily apparent based on the key adaptations (or lack thereof) observed in the present study. It is possible that undertaking performance testing in the same environment that has been experienced during training could provide some psychophysiological benefit compared to control (temperate training only). Moreover, non-haematological benefits such as improved buffering capacity, reduced sweat rates, exercise efficiency and others not measured in the current study but noted elsewhere [26,37] may also contribute to improved performance in both environmental exposure groups compared to control. Indeed, both intervention groups displayed higher lactate levels post 15 min time trial compared to control, possibly indicating both groups were able to exercise (or “push” themselves) harder during the post-intervention trial compared to control.

Plasma volume expansion is a primary adaptation observed early in the heat acclimation process [38], whereas PV is reduced following hypoxic exposure [39]. The magnitude of change in PV can sometimes vary greatly due to differences in the participants’ fitness, temperature and total hypoxic stimulus and hydration status [40]. Another source of non-uniform findings is the different methods employed to measure PV changes over the period of environmental exposure. While method comparisons are rare, direct methods such as the CO-rebreathing technique that measures total circulating haemoglobin have proven relatively sensitive and precise in determining blood volumes, and thus are preferred [40]. As explained in the Methods section, we originally planned to use the CO-rebreathing technique. Due to unforeseen technical issues, we were unable to undertake the technique post training and thus relied on the indirect methods based on blood constituents such Hct and (Hb) to estimate PV changes described elsewhere [35]. Although many studies have used the Dill and Costill method for calculating changes in PV, and its application has been deemed appropriate for calculation of plasma and serum biomarkers only [41], we recognise that such indirect methods have their limitations. It is understood that results using the indirect method can be affected by changes in the blood constituents or shrinkage or swelling of erythrocytes, thus causing over- or underestimated changes in PV. Further, while the technical error of measurement for (Hb) and Hct appear comparable to direct methods [42], we understand that their use in a calculation to estimate PV could lead to much larger error than the direct methods. Therefore, the calculation methods employed in the current study are an estimate only, and our findings and thus interpretation of such are purely speculative and do not permit us to draw firm conclusions regarding the exact magnitude of PV changes.

Notwithstanding, our data demonstrated a 3.8% increase in PV in the heat acclimation group, which is similar to McCleave’s study (~3.8%) [31], but slightly less than reported by other studies (+5–6%) [8,32]. Larger responses observed by other studies could be due to the length or type of heat exposure, with participants in the Buchheit study undertaking skills sessions in hot environments (approx. 1 h a day), whereas participants in the Rendell study completed a more controlled isothermal heat strain session for 90 min [8,29]. The heat sessions in the McCleave study were intermittent (3 times per week) over a 3-week period, and while practical in nature, direct comparisons to our daily heat exposure are difficult to make. Nevertheless, it is well known that increases in PV (and blood volume) reduce the burden on the cardiovascular system to simultaneously meet muscular blood supply for metabolic demands, and skin blood flow during exercise in hot conditions [41]. In the present study, enhanced thermoregulation could explain the attenuation of temperature increases in the heat-acclimation group compared to heat acclimation with hypoxic exposure after the initial 75 min steady-state component of the exercise protocol where the metabolic demands of exercise are similar. This was supported in the metabolic (blood lactate), and physiological (heart rate) stress markers which, although not significant, were also cumulatively attenuated in the heat acclimation compared to heat acclimation with hypoxic exposure.

In contrast, heat acclimation with hypoxic exposure resulted in a reduction in PV (4.1%) following the 12-day intervention period which contradicts some previous works that demonstrated an increase [8,26,28]. McCleave et al. [31] demonstrated little change in PV following 21 days of both heat and hypoxia, suggesting PV expansion was potentially blunted by a higher hypoxic stimulus. When comparing total hypoxic dose based on the equation by Garvican-Lewis et al. [43] our dose (~360 km·h) is much lower than McCleave et al.’s [31] dose (~819 km·h), but somewhat comparable to others that showed an increase in PV (~260 km·h [8] and ~529 km·h [29]). Whether the heat dose was not sufficient to overcome the hypoxic-induced haemoconcentration is unclear, but unlikely given that the heat dose used in the present study was similar to others [8,31]. It is highly probable that the reductions in PV observed in the combined heat and hypoxia group were a reflection of large total-body water loss as a result of hypoxic-induced diuresis (evidenced by a significant decrease in body weight (~1.7%) in that group only) [36]. This has been previously reported following acute exposure to high altitude, and we hypothesise that intermittent cyclical hypoxic exposure in the current study may have triggered diuresis in our participants, and if not immediately replaced by ingested water, reductions in body mass can occur [44]. Blood and weight were taken in the morning after exiting the altitude chamber, and fluid intake was not monitored overnight. Therefore, we are unable to determine if the decrease in PV is a reflection of acute changes in water balance or that physiological mechanisms involved in water balance at altitude (or with hypoxic exposure) over the 12 days is overwhelming, possibly negating the positive heat adaptations when combined. This concept requires further investigation but highlights an important practical consideration when implementing LHTL and for coaches/trainers to closely monitor body-weight changes and ensure appropriate fluids are given to replace water loss.

The decrease in total body-mass in the heat acclimation and hypoxia group was ac-companied by an increase in (Hb) (~2.9%), leading to haemoconcentration. Hb mass, usually measured by the CO-rebreathing technique, has been shown to increase in elite cyclists following 12 days of LHTL [45]. However, more current evidence suggests longer exposure periods (3–4 weeks) at altitudes of ~2500 m are required to yield increases in red cell mass [28,45]. Thus, the changes in (Hb) following 2 weeks of heat acclimation with or without hypoxia are most likely a reflection of acute changes in plasma volume rather than changes in erythropoietin levels and Hb mass. The shorter exposure period chosen in the present study was of practical rationale for ‘semi-professional’ athletes trying to implement these potential performance-enhancing strategies in the lead-up to competition. Living at altitude is not available to all athletes and LHTL can be very costly and time-consuming for these individuals. Moreover, hypoxic facilities/tents are not sustainable as an 18–24 h conditioning procedure. Thus, the present study wanted to examine the short-term changes in haemodynamic responses to environmental challenges as well as (not exclusively) haematopoietic factors using a time period that was practically relevant to our ‘semi-professional’ athletes.

The current study design has several sports-specific strengths for competitive athletes. Firstly, our study used a homogenous group of well-trained athletes preparing for competition in hot, humid conditions, and we wanted to replicate a typical preparation undertaken by these athletes. We instructed the athletes to maintain their normal training volume and intensities in addition to performing daily heat-training sessions over the 2 weeks. Secondly, our study used a closed-end performance test with high ecological validity (i.e., 15 min time-trial for distance test), and thus outcomes from this test, from an athletes’ perspective, have high relevance in a competition setting [46].

The present study is also not without its limitations. Skin-temperature measures were not included, and core temperature was extrapolated using tympanic temperatures. This measure, taken by the same operator throughout the study, was duplicated to reduce variability. The increases with exercise and heat exposure in the present study were similar to rectal temperature changes previously observed [16,47]. As discussed previously, red cell mass measurements using CO-rebreathing were unsuccessful post-environmental treatment due to equipment and technical limitations and secondary measures (e.g., (Hb) and Hct) were used to approximate the hypoxic adaptations. Though the addition of a control group is the strength of the study design, they did not specifically complete the same daily environmental rides in thermoneutral ‘lab’ conditions as the other groups. However, major outcomes of this group were stable without heat exposure. Given the ‘open-label’ study design of the study, all participants were aware of the designated intervention group. Thus, we cannot discount the possibility of either a placebo or training effect in either or both intervention groups. Finally, we were unable to control for menstrual cycle phase but given a fairly even spread of females across groups (H+NH:2, H:1, CON:1), and individual female data showing minimal differences from the group mean, it is unlikely to significantly impact the key findings.

## 5. Conclusions

Based on the results of this study and others, there appears to be little evidence that the addition of hypoxic exposure to heat acclimation provides any further benefit to short-duration exercise performance when evaluated immediately following an exposure period. Given most performance outcomes in prior studies were evaluated in temperate environments, we extended these findings with application in a hot, humid environment, which is of practical relevance to athletes. Interestingly, despite divergent adaptations when hypoxia was added to heat acclimation, an ergogenic effect was still observed. However, given the limitations in our haematological methods, it is difficult to clearly identify the mechanisms underpinning the ergogenic effect of adaptation to heat (with, or without hypoxia), but importantly, caution should be used when interpreting our haematological adaptation findings. Future research is needed to further explore the sequence of the combined protocols, timing and total exposure amount of heat and hypoxic interventions relative to competition phases to confirm performance-enhancing opportunities for well-trained and elite endurance athletes.

## Figures and Tables

**Figure 1 sports-10-00069-f001:**
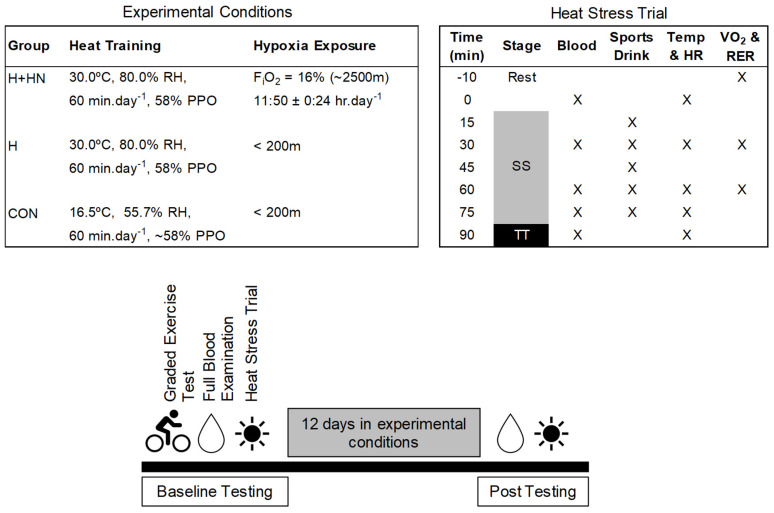
Trial day sequence of events and measurements. Temperature, heart rate (HR), rating of perceived exertion (RPE), oxygen consumption (VO_2_), and respiratory exchange ratios (RER) were obtained at rest and during exercise. Blood samples were collected at rest, throughout exercise, and during recovery. Abbreviations: SS-steady state; TT-time trial; PPO-peak power output; RH-relative humidity and FiO_2_-fraction of inspired oxygen.

**Figure 2 sports-10-00069-f002:**
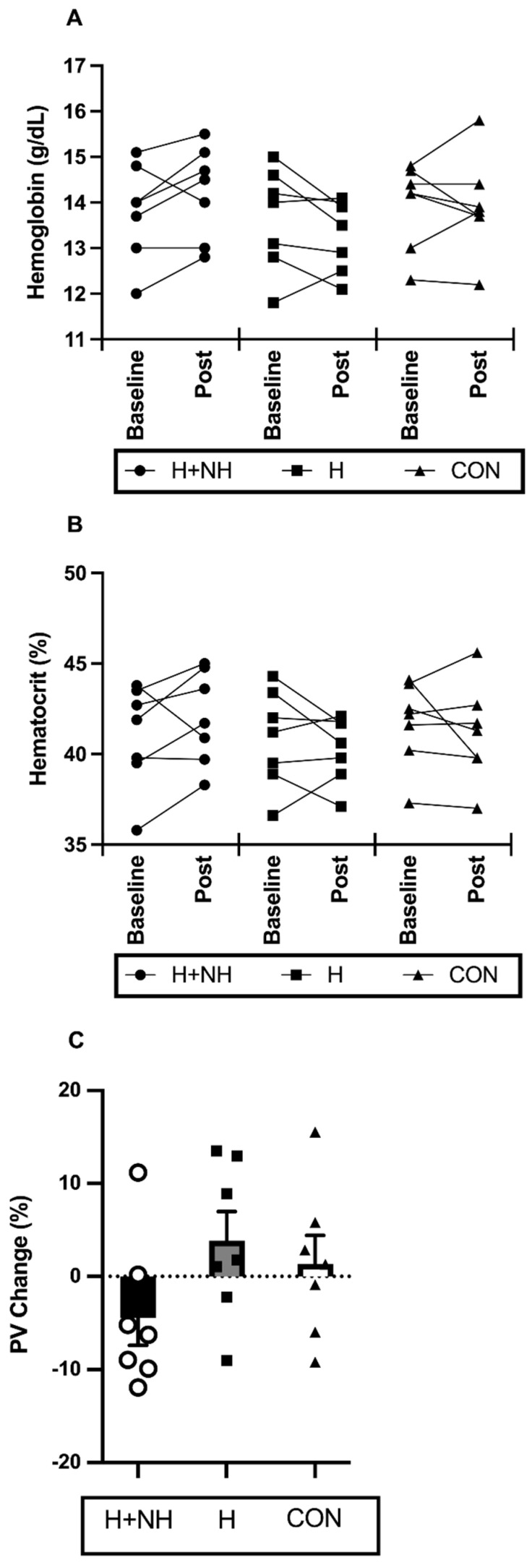
The absolute change in individual (**A**) haemoglobin and (**B**) haematocrit levels and percentage change in individual and average (**C**) plasma volume (PV) at before (baseline) and/or after interventions (post).

**Figure 3 sports-10-00069-f003:**
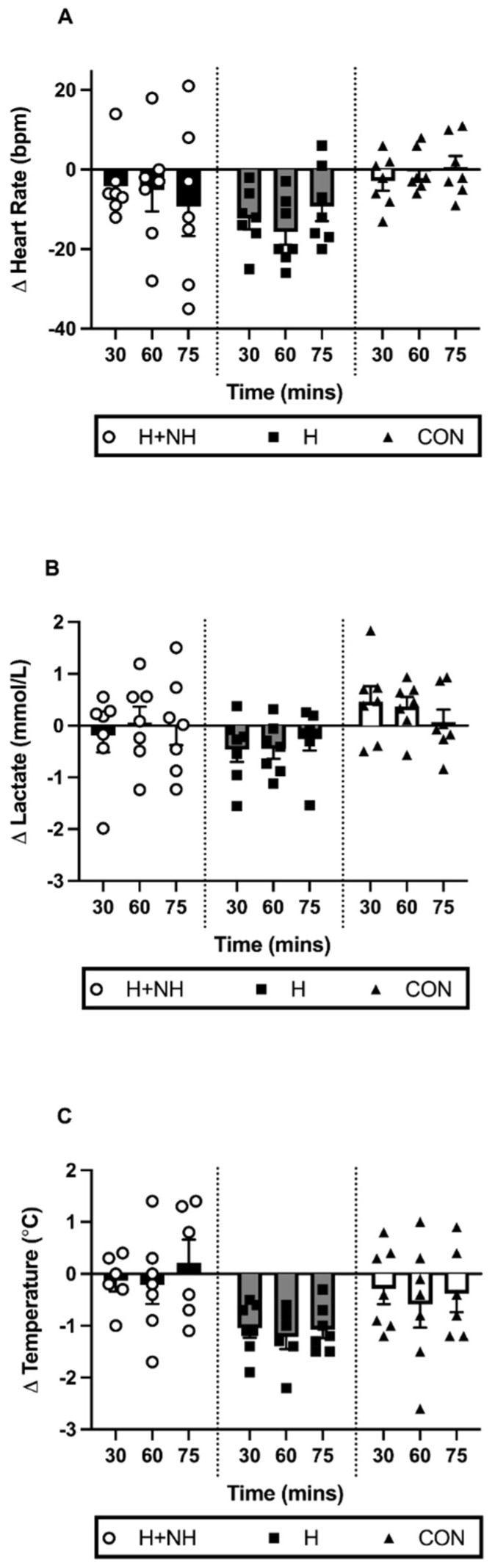
The change in individual and average (**A**) heart rate; (**B**) blood lactate; and (**C**) tympanic temperature levels during steady state between the baseline and post-intervention heat trials.

**Figure 4 sports-10-00069-f004:**
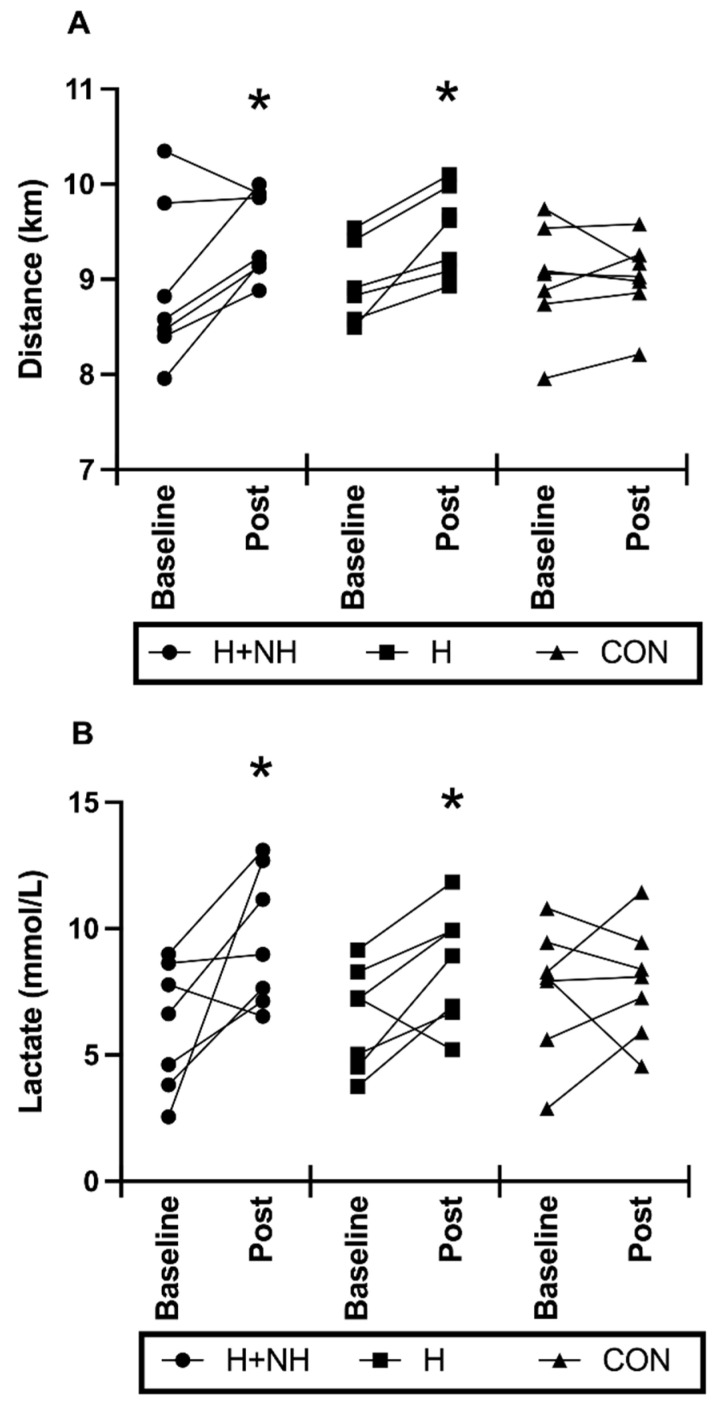
Individual and average values for (**A**) Total distance covered during the 15 min maximal effort time trial portion; (**B**) peak lactate levels following time-trial portion before (baseline) and after (post) heat exposure. * Significantly different to baseline (*p* < 0.05).

**Table 1 sports-10-00069-t001:** Participants’ physical characteristics, training loads and environment exposure duration.

	H+NH	H	CON	*p* Value
N (Male/Female)	7 (5/2)	7 (6/1)	7 (6/1)	-
Age (years)	32 ± 2	32 ± 4	39 ± 2	0.144
Height (cm)	175.8 ± 3.2	178.5 ± 1.5	180.0 ± 2.5	0.497
Weight (kg)	70.0 ± 2.4	73.1 ± 2.9	78.4 ± 3.8	0.180
Absolute VO_2max_ (L·min^−1^)	4.3 ± 0.4	4.6 ± 0.3	4.6 ± 0.3	0.269
Relative VO_2max_ (mL·kg·min^−1^)	61.8 ± 3.3	62.6 ± 2.2	58.9 ± 2.9	0.503
VO_2max_ peak watts	344 ± 23	357 ± 10	364 ± 18	0.300
VO_2max_ peak watts.kg^−1^	4.9 ± 0.2	4.9 ± 0.2	4.6 ± 0.2	0.454
Trial workload ^1^ (W)	221 ± 13	222 ± 7	225 ± 12	0.970
% of peak watts	62.9 ± 0.1	62.0 ± 0.5	62.2 ± 0.8	0.554
Daily heat training workload ^2^ (W)	199 ± 13	205 ± 7	-	0.728
Hypoxia exposure ^3^ (h:min:s)	11:50:00 ± 0:24	-	-	-

^1^ Trial workload is the workload completed during the steady-state (75′) portion of the heat-stress trial; ^2^ daily heat training workload is the average training workload for both H and H+NH groups over the 12 days; ^3^ hypoxia exposure is the average total exposure experienced by participants over the 12 days.

**Table 2 sports-10-00069-t002:** External self-training total exposure, percentage breakdown of exercise modality and average reported intensity over the 12-week intervention period.

	H+NH	H	CON	*p* Value
External training exposure ^1^ (avg. min)	2616 ± 214	2751 ± 157	2233 ± 453	0.474
Swimming/Cycling/Running (%)	15/42/21	26/42/30	11/58/18	
Other (incl. gym, Pilates) (%)	21	2	13	
Average reported intensity				
Swimming (m/min)	51 ± 7.0	49 ± 4.6	47 ± 7.0	0.690
Cycling (km/h)	30 ± 2.4	29 ± 4.3	27 ± 2.4	0.873
Running (min/km)	4.9 ± 0.2	4.7 ± 0.2	5.0 ± 0.2	0.424

^1^ External training exposure is the average total amount in minutes of training performed in addition to the 60 min of daily training for all 3 groups during the 12 days.

**Table 3 sports-10-00069-t003:** Baseline physiological responses to submaximal and maximal exercise in heat.

								*p* Values		
			Submaximal Exercise			Maximal Exercise		Interaction	Within	Between
		Baseline	30′	60′	75′	90′			Group ^1^	Group ^2^
VO_2_			_a_	_a_						
(L·min^−1^)	H+NH	0.28 ± 0.02	3.19 ± 0.13	3.17 ± 0.16	-	-	3 Grp	0.307	<0.001	0.276
	H	0.36 ± 0.03	3.55 ± 0.14	3.58 ± 0.17	-	-	2 Grp	0.117	<0.001	0.090
	CON	0.36 ± 0.02	3.50 ± 0.15	3.55 ± 0.17	-	-				
RER			_a_	_a_						
	H+NH	0.84 ± 0.02	0.89 ± 0.01	0.89 ± 0.01	-	-	3 Grp	0.236	0.014	0.173
	H	0.83 ± 0.04	0.84 ± 0.02	0.84 ± 0.01	-	-	2 Grp	0.499	0.246	0.139
	CON	0.79 ± 0.02	0.87 ± 0.01	0.88 ± 0.01	-	-				
RPE										
	H+NH	-	13.9 ± 0.5	15.2 ± 0.6	15.6 ± 0.9	-	3 Grp	0.413	0.209	0.168
	H	-	14.3 ± 0.8	16.1 ± 0.4	16.6 ± 0.7	-	2 Grp	0.374	0.564	0.116
	CON	-	13.0 ± 0.5	15.4 ± 0.6	15.9 ± 0.8	-				
Temperature			_a_	_a_	_a_	_a,b,d_				
(°C)	H+NH	36.2 ± 0.3	37.4 ± 0.1	37.6 ± 0.2	37.3 ± 0.4	38.0 ± 0.3	3 Grp	0.984	<0.001	0.251
	H	36.3 ± 0.1	37.8 ± 0.1	38.1 ± 0.2	37.9 ± 0.3	38.3 ± 0.3	2 Grp	0.882	<0.001	0.119
	CON	36.2 ± 0.2	37.5 ± 0.2	38.0 ± 0.3	37.8 ± 0.3	38.0 ± 0.3				
Heart Rate			_a_	_a_	_a_	_a,b,c,d_				
(bpm)	H+NH	54 ± 3	159 ± 4	165 ± 5	169 ± 4	183 ± 4	3 Grp	0.624	<0.001	0.675
	H	49 ± 2	157 ± 7	162 ± 6	156 ± 7	177 ± 8	2 Grp	0.454	<0.001	0.423
	CON	52 ± 3	153 ± 3	159 ± 3	159 ± 4	174 ± 4				
Lactate			_a_	_a_	_a_	_a,b,c,d_				
(mmol·L^−1^)	H+NH	0.89 ± 0.06	2.27 ± 0.34	2.23 ± 0.28	2.48 ± 0.31	7.48 ± 1.27	3 Grp	0.994	<0.001	0.450
	H	1.08 ± 0.11	2.41 ± 0.30	2.60 ± 0.34	2.36 ± 0.37	7.97 ± 0.88	2 Grp	0.986	<0.001	0.514
	CON	1.04 ± 0.11	2.54 ± 0.31	2.70 ± 0.36	3.14 ± 0.50	8.43 ± 0.86				

^1^*p* values for comparisons across all time points, main effect of time; ^2^
*p* values for comparisons between groups (H+NH vs. H vs. CON for 3 Grp; H+NH vs. H for 2 Grp). _a_ significantly different from baseline (*p* < 0.05); _b_ significantly different from 30′ (*p* < 0.05); _c_ significantly different from 60′ (*p* < 0.05); _d_ significantly different from 75′ (*p* < 0.05).

**Table 4 sports-10-00069-t004:** Body mass before and after interventions and heat trial.

							*p* Values			
		Baseline Trial		Post Intervention Trial			Interaction	Within	BTW	BTW
		Rest	Post	Rest	Post			Group ^1^	Group ^2^	Trial ^3^
Body Mass										
(kg)	H+NH	70.2 ± 2.3	70.0 ± 2.4	68.9 ± 2.1 *	68.3 ± 2.1 #,*	3 Grp	0.049	0.002	0.170	0.019
	H	73.9 ± 3.1	73.2 ± 3.0 #	73.9 ± 3.3	73.1 ± 3.2 #	2 Grp	0.020	0.003	0.299	0.018
	CON	78.6 ± 3.9	78.1 ± 3.8 #	78.1 ± 3.9	77.7 ± 3.9 #					

^1^*p* values for comparisons between resting and post-exercise time points within the same trial; ^2^
*p* values for comparisons between intervention groups (H+NH vs. H vs. CON for 3 Grp; H+NH vs. H for 2 Grp); ^3^
*p* values for comparisons between baseline and post trial; # significantly different from rest value (*p* < 0.01); * significantly different from baseline trial (*p* < 0.05).

## Data Availability

The datasets generated during and/or analysed during the current study are available from the corresponding author on reasonable request.

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
