# Peer review of "Heat Acclimation with or without Normobaric Hypoxia Exposure Leads to Similar Improvements in Endurance Performance in the Heat"

_sports, 2022, doi:10.3390/sports10050069_

Round 1
Reviewer 1 Report
This manuscript addresses a very relevant area for athletes using environmental interventions in their performance endeavours. I have some methodological concerns about the physiological measures used in the study, mainly around the techniques used for blood volume assessments. This is briefly discussed in your limitations, but I think this needs a lot more attention throughout the manuscript, to frame the context around your results. I have more specific comments on this below.
While I think the methodology used in some of the physiological measures makes strong inferences relating to those adaptations difficult, there are still interesting data around the performance changes with all conditions. However, more work is needed to describe control and monitoring around the training process during the study. I am a little confused about changes to the protocol for participants who had upcoming races, and the reporting around the control group training and on-supervised sessions (for all groups) currently lacks detail.
I think if some more detail can be added around the methods, and much more discussion around the technical error of measurements and limitations with the physiological testing (most notably blood volumes) this work would be a good contribution to this topic area.
Specific comments:
Line 18: the H+A abbreviation doesn’t quite make sense here. I am assuming the A is representing altitude, but you have not described the hypoxia as simulated altitude (or similar), so this could be confusing for people to read, in this format
Line 20: it would be helpful to mention the exercise mode here
Line 25: I think adding the absolute time trial numbers is relevant here, or in the methods – it is hard to understand a 5.5% change without knowing how long the time trail was, overall. If reporting distance, the exercise mode would also be important here, as per my previous comment
Line 39: the opening to this sentence doesn’t seem like a good description of LHTL – the insertion of LHTL seems out of place here
Line 47: I think the words ‘for a given work rate’ should come before ‘maximal cardiac output’ in this sentence, for it to be completely correct
Line 48: this sentence is a little incomplete – is Hypervolemia an adaptation to heat training? that is not mentioned when introducing this sentence, or in previous sentences
Line 52: are these adaptations evident following heat acclimation?
Line 59-60: muscle buffering capacity could also be relevant to mention here
Line 61: iI think this sentence is incorrect – enhanced muscle and mitochondrial efficiency does not increase maximal oxygen consumption. There are also many studies showing improved performance following hypoxic exposure without concurrent improvement in maximal oxygen consumption, which is not really acknowledged in the way this sentence is currently written
Line 65: is this longer durations comment specific to altitude exposure?
Line 101: what was the inclusion criteria here, to ensure/define ‘well-trained’?
Line 113: what were the participants told about the study design before participation? For example, did the control group know that the study was examining heat and hypoxic interventions? Similarly, did the heat only group know that there was another H+A group they were being compared to? This is important information to understand any possible placebo/nocebo effects
Line 178: what is the technical error of measurement using this technique to estimate PV changes?
There is a methodological issue here with your blood measurements that needs to be addressed. It appears as throughout you have measured only venous blood samples, and in particular, extrapolated a lot of your findings from hemoglobin concentration. This is an issue given the sensitivity of this measure in the context of exercise capacity (as an example, see Otto et al ref. below) and extrapolation of PV from Hb concentration and Hct [as opposed to extrapolation from CO rebreathing (used in many of the studies you reference) or an Evans blue dye technique].
I know this limitation receives one line of attention in your limitations section, but given that these blood volume measures are such a strong part of your aims, the sensitivity and technical error of the eventual techniques you used needs to be discussed in more detail, to frame your results.
From reading your limitations section, it also seems that CO rebreathing was part of your methods – but this does not appear in the methods section. I think it would be more accurate to present this as one of your methods and discuss the ‘unsuccessful post-environmental treatment due to equipment and technical limitations’ in your results. This makes more sense from a study design perspective, and is a valuable piece of information (if more details are provided) for other researchers seeking to complete similar work).
Otto, J. M., Plumb, J. O., Wakeham, D., Clissold, E., Loughney, L., Schmidt, W., ... & Richards, T. (2017). Total haemoglobin mass, but not haemoglobin concentration, is associated with preoperative cardiopulmonary exercise testing-derived oxygen-consumption variables. BJA: British Journal of Anaesthesia, 118(5), 747-754.
Line 189: how many participants fell into this cater=gory? I am a little confused about the overall control of the training program, outside the 12 x 60 min sessions. What did the control group do for these 12 sessions?
Line 199: what were the dimensions of the rooms the participants were staying in? There are some suggestions that reduced ambulation in these simulated environments may have an effect, so I think providing this detail is important
Line 213: what do you mean by ‘average sum’? Also, I think minutes would be better described as exposure, as this does not actually account for any load completed (i.e. theoretically, if the intensity is nearing rest, the load could be very minimal)
Line 226: stating the participants ‘were well trained’ is a very subjective comment (particularly in the absence of any definition for this qualitative descriptor) and I think is inappropriate in the results section. I think you should stick to reporting the result here (i.e. mean VO2max)
Table 1: what is ‘trail workload’?
Table 1: I think ‘External training load’ would be better described as ‘training exposure’. Also, I think this needs to be reported as a breakdown of swimming, running, cycling and gym work, as they will all have different training adaptations. Also, is this training duration across the 12 days of the study (I have the same question for hypoxia exposure)?
Line 251: Hb had been previously defined, but [Hb] has not been. Is this hemoglobin concentration?
Line 253: why have you not presented calculated plasma volume data in Table 3? I think this data should be presented somewhere (in a table or figure), given it is one of the main physiological measurements you are discussion
Figure 2: what is (hmp) on the y-axis in panel A
Line 281: I think including effect sizes throughout would really help the interpretation of your results. You discuss a number of ‘trends’ but it is hard to put these into context. I also think effects sizes would be a much better method of discussing changes, as compared to some of the percentage changes you cite throughout
Line 298-299: I don’t think this sentence is quite correct – your primary measure of thermoregulation i.e. tympanic temperature, was clearly different in the H group.
Line 334-337: the methods used need some discussion here – most of the studies you references here are using the CO rebreathing technique, and it is not clear the typical error that should be expected with the venous [Hb] and Hct sampling technique (and subsequent calculations used in your study. A lot of the CO studies you cite report PV CV’s of around 4% using the CO rebreathing technique. Given that it is likely there is additional error in your methods, this methodological limitation needs more discussion and attention.
Line 377-383: I think it is important to acknowledge here that both groups had the same improvement ins exercise performance. Therefore, despite apparent physiological differences, the same performance outcome was achieved. This has been seen in some studies and contrasts with others, so clearly articulating this in your findings helps to contribute to the (sometimes confusing) body of knowledge here.
Line 391: I am not quite following this point – wouldn’t a fixed amount of work be more representative of racing, compared with a fixed time to complete as much work/distance as possible?
Line 398: COP not previously defined. It might also be worth inserting a reference about this technique, as you have not previously discussed it. Although, as per my previous comments, I think this technique should be discussed earlier in the manuscript
Line 407-409: it would be helpful to mention the duration of maximal exercise that this might apply to – as it reads right now, this statement is very broad (i.e. applies to all maximal exercise)
Line 413: using the language ‘not significantly hindered’ is very directionally biased language. You could also say that ‘exercise performance was not significantly improved'. I think more neutral language, such as ‘was similar’, is far more appropriate and less biased. There are also instances of this type of language earlier in the manuscript.
Reviewer 2 Report
Overview:
In this study, Hanson et al. investigated whether there is an additive effect of exposure to both heat and hypoxia for 12 consecutive days compared to heat exposure alone and a control group. The authors should be commended for recruiting 21 highly trained subjects for a 12-day intervention study. While both intervention groups improved time trial performance, there were no significant differences between groups. Differences between groups from pre to post intervention were noted in PV and [Hb], and tympanic temperature during a steady state cycling trial.
Major comments:
There are two major concerns with this study:
- First, in the introduction, the authors propose that 150 hrs of hypoxia have positive effects on endurance performance and cite a study from >20 years ago. Based on extensive, and more current literature, 12 days of 12hr/day of hypoxic exposure are very likely insufficient to induce hematological adaptations that would result in improved performance. Furthermore, the hypoxic dose implemented in the study design may have only led to negative outcomes (substantial reductions in HCT), while not being sufficient for actual haematological adaptions (only a pseudo-increase in [Hb] due to reduced PV). Thus, it can only be concluded that short-term hypoxic exposure combined with heat acclimation may not pose an additional beneficial effect on performance compared to heat alone. This should be addressed.
- [Hb] is not an appropriate indicator of haematological adaptations following heat and or hypoxic exposure as it is dependent on PV, which is affected by both interventions used in the study. The impact of changes in PV on [Hb] is clearly shown in this study and therefore this parameter may not be an appropriate indicator of hematological adaptations following heat and/or hypoxic exposure. This is only very briefly mentioned in the discussion. I appreciate that there was an attempt to measure Hb-mass and that the authors faced problems with this measurement, however the problematic nature of [Hb] should be addressed and acknowledged at an earlier stage and not only in the limitations section.
Minor comments:
Intro
- Lines 48-49: Does hypervolemia (especially that brought about by heat acclimation) cause a reduction in sweat rate? I believe it should be the opposite (e.g., Periard et al., SJMSS, 2015 doi: 10.1111/sms.12408)
- Lines 63-65: please check (and cite) more current literature as it is known that more than 150 hrs of hypoxia are for significant hematological adaptations that would translate to performance improvements.
Methods
- Line 146 – the reference does not seem to match the text. Also, it is not mentioned that lactate was measured during the VO2max test in the relevant paragraph, or how it was measured throughout the study.
- It is not clear why venous blood samples were taken repeatedly during the heat stress trial. What markers were assessed (data are missing from the manuscript) and how were the bloods analyzed for these markers?
- Was SaO2 monitored during the study in the H+A group?
Results
- Overall, the results section needs to be written more clearly as it is extremely hard to follow and understand the comparisons made.
- Please provide details about the specific training undertaken by the athletes and comparisons between groups (e.g., in terms of volume and intensity).
- 3 female athletes were included in the study. If the pre/post intervention tests were conducted ~12 days apart, these female athletes were very likely tested at two distinct phases of their menstrual cycle, which could affect some of the outcome measures (e.g. [Hb], PV, performance, etc.) independently of the intervention introduced. Was this taken into account? Did the female participants show different trends in the outcome measures assessed compared to males? As much as I am all about including female subject in sport science research, this needs to be done appropriately (for example, as much as possible ensuring an equal number of male and female subjects overall in the study, and even more importantly, and equal number of females in each study group).
- Table 3 – consider showing in figure format. Also, drink volume does not seem relevant and can be removed or mentioned in text.
- Section 3.3
- Were the differences in body mass observed at a given time point or was the delta (∆) change different?
- This paragraph needs to be worded more clearly as it is quite confusing which remarks relate to pre-post heat stress trial and which to pre-post 12-days of intervention.
- Sections 3.4
- The phrase “post exposure” is confusing as one of the groups was not exposed to anything and the two other groups had different exposures… (same in the legend of figure 2).
- Were differences between groups in lactate and temperature compared as absolute values at each time point or as ∆ change, as shown in the figures? If the former, it is not appropriate to show significance symbols in figure 2.
- Figure 2 – use standard units for HR. Also, it seems odd that HR for some participants was reduced by > beats from pre to post intervention. How do you explain this exaggerated response?
- Figure 3 – it might be more relevant to show individual changes from baseline to post intervention with a line representing each participant so that individual responses can be visualized and determined (and not only averages and spreads).
Discussion
- Results from the current study should not be compared to previous literature utilizing intermittent hypoxia, hypoxic training, etc. (e.g., McLeave IJSPP, 2020)
- Line 343 – if fitness level may have affected changes in PV, why would this occur only in the H+A group and not H group?
Round 2
Reviewer 1 Report
I think the authors did a really good job of addressing the reviewers comments (it's clear you really took this process seriously, which is really nice to see when everyone puts so much time into these reviews).
Many of my original comments are now much clear for me to uns=derstaind and I doing;t have any other comments that are going to meaningfully contribute.
Nice work, and thanks for engaging with my review as you did - I enjoyed reading and discussing your work with you.
Reviewer 2 Report
The Authors have adequately addressed my comments and the I believe the manuscript has greatly improved from its first version. Well done and good luck.